# LLMatic: Neural Architecture Search via Large Language Models and Quality Diversity Optimization

## Abstract

Large Language Models (LLMs) have emerged as powerful tools capable of accomplishing a broad spectrum of tasks. Their abilities span numerous areas, and one area where they have made a significant impact is in the domain of code generation. Here, we propose to use the coding abilities of LLMs to introduce meaningful variations to code defining neural networks. Meanwhile, Quality-Diversity (QD) algorithms are known to discover diverse and robust solutions. By merging the code-generating abilities of LLMs with the diversity and robustness of QD solutions, we introduce `LLMatic`, a Neural Architecture Search (NAS) algorithm. While LLMs struggle to conduct NAS directly through prompts, `LLMatic` uses a procedural approach, leveraging QD for prompts and network architecture to create diverse and high-performing networks. We test `LLMatic` on the CIFAR-10 and NAS-bench-201 benchmark, demonstrating that it can produce competitive networks while evaluating just $2,000$ candidates, even without prior knowledge of the benchmark domain or exposure to any previous top-performing models for the benchmark. The open-sourced code is available in `github.com/xxxx`.

## 1 Introduction

A major challenge in deep learning is designing good neural network architectures. Neural Architecture Search (NAS) is the generic term for various approaches to automating this design process White et al. (2023). The idea is to formulate an objective, such as maximum accuracy on a classification problem with a given budget of parameters and training cycles, and cast the problem as a search for the architecture that maximizes the objective. This typically means that many thousands of architectures are tested and discarded in the process. Every test consists of training the candidate network architecture using some form of gradient descent on the chosen benchmark dataset to measure its performance.

Two common algorithmic approaches to NAS are reinforcement learning and evolutionary computation. Reinforcement learning approaches to NAS (Jaafra et al., 2019) train a controller (typically another neural network) that outputs network architectures; these network architectures are tested and their performance is used as a reward signal. Evolutionary computation approaches to NAS (Liu et al., 2021), on the other hand, directly search the space of neural architectures. A population of architectures are kept, and their performance is used as a fitness score. Evolutionary NAS approaches are similar to neuroevolution, which has existed since the 1980s (Tenorio & Lee, 1988; Miller et al., 1989), and one might even see NAS as a form of neuroevolution. The main difference is that in NAS, the search process does not concern the parameters of the neural network, only its architecture.

One could argue that search by evolutionary computation or reinforcement learning is quite mindless and wasteful, given how many architectures need to be tested and how uninformed the changes that lead to each new architecture are. Is there some way we can inform the search by exploiting stored knowledge about how to design neural networks? This paper explores the idea that we can do exactly this using code-generating large language models (LLMs). More precisely, we propose using LLMs to generate new architectural variations.

The argument for this is simply that modern LLMs fine-tuned on code are very capable. Given the amount of machine learning code they have been trained on, it is not surprising that they can

design good neural network architectures. However, an LLM by itself cannot in general find an optimal architecture for a given problem, as it cannot test architectures and learn from its experiments. Therefore, we propose combining the domain knowledge of code-generating LLMs with a robust search mechanism.

While generating a single architecture that maximizes a given objective is good for many use cases, there is in general more value to generating a set of architectures that vary across some relevant dimensions. For example, one might want to have a set of architectures that vary in their parameter counts or depths. This helps in understanding the trade-offs between various desirable metrics and could assist in making better-informed decisions about which architecture to use for a specific application. For example, one might want a range of networks for edge deployments to clients with different RAM sizes. To enable this, the solution proposed here leverages quality-diversity search Pugh et al. (2016), specifically a version of the MAP-Elites algorithm (Mouret & Clune, 2015).

Our main contribution is a novel LLM-based NAS algorithm, `LLMatic`1, that utilizes the power of two QD archives to search for competitive networks with just 2000 searches. We empirically show the performance of `LLMatic` on the CIFAR-10 dataset and the NAS-bench-201 benchmark where `LLMatic` searches for networks with performance near to state-of-the-art results.

## 2 RELATED WORK

Designing good, learnable neural architectures can be an expensive and unintuitive process for human designers. Neural Architecture Search (NAS) aims to automatically find neural architectures capable of strong performance after training (Elsken et al., 2019). Bayesian methods are a popular choice given their low sample complexity and the fact that evaluating each architecture (by training it) can be computationally expensive (Kandasamy et al., 2018). Alternatively, reinforcement learning can be used to train an agent (usually another neural network) to output candidate architectures for a given task, with the performance after training of the candidate architecture acting as a reward signal (Jaafra et al., 2019). Evolutionary methods can also be used to search directly through the space of possible architectures (Liu et al., 2021). Similarly, Monte Carlo Tree Search has been used to search the space of possible architectures (Wistuba, 2017). In all cases, a human designer must manually define a set of atomic network components or edit actions for use in network search/generation.

To avoid having the designer constrain the space of possible architectures prior to search, we turn to code-generating Large Language Models (LLMs), large models trained auto-regressively on massive datasets of code (e.g. public repositories in Github). Transformers (Vaswani et al., 2017) facilitated the explosion of LLMs (Radford et al., 2019; Brown et al., 2020). Apart from creating state-of-the-art models in core natural language processing tasks (Adelani et al., 2022; Nasir & Mchechesi, 2022), they led to creating models for a wide variety of other tasks, such as generating video game levels and code (Chen et al., 2021; Todd et al., 2023; Nasir & Togelius, 2023).

Recently, LLMs have been used for evolving code by curating it as an evolutionary problem. Evolution through Large Models (ELM) (Lehman et al., 2022) has introduced evolutionary operators through LLMs and MAP-Elites (Mouret & Clune, 2015) to evolve robots morphology at a code level. We take our inspiration from ELM. EvoPrompting (Chen et al., 2023) is an LLM-based method that is somewhat similar to ours in that it uses code-LLMs as mutation and crossover operators in order to perform NAS. It is tested on the MNIST-1D classification task (Greydanus, 2020) and the CLRS algorithmic reasoning benchmark (Veličković et al., 2022). Since performance can generally be trivially increased by simply adding parameters to the model, an additional penalty is added to the fitness of a candidate neural architecture corresponding to its model size. This favours small models with effective architectures. In our method, we instead consider model complexity (in terms of FLOPS) as a diversity metric, searching for high-performing models of a variety of sizes. GENIUS (Zheng et al., 2023) is another LLM-based NAS algorithm that uses GPT-4 to simply search through straight-forward prompting.

Quality Diversity (QD) methods (Pugh et al., 2016) are a family of evolutionary algorithms that, in addition to optimizing a fitness metric, search for a diversity of individuals according to some user-specified "behavioral descriptors". Instead of keeping a population of the most fit individuals, QD methods such as MAP-Elites (Mouret & Clune, 2015) maintain an "archive" of individuals,

where this archive is partitioned into cells, with each cell corresponding to individuals exhibiting a particular range of values along each behavioral descriptor.

QD methods are valuable in domains such as robot control, where it is useful to learn diverse high-quality trajectories, in case one solution should become unavailable during deployment because of a physical obstruction or mechanical malfunction. Another motivating factor is that greedily searching for the fittest individual may not be desirable in deceptive domains. Here, maintaining a diversity of fit individuals may protect the population from falling into local optima. Conversely, diverse, unorthodox solutions may provide valuable "stepping stones" on the path to globally fit individuals.

## 3 APPROACH

---

**Algorithm 1:** LLMatic

**initialize :** network_archive, prompt_archive, best_loss, initial_network;
selected_network = initial_network;
**foreach** *Generation_i in Generations* **do**
    **if** *len of archives < num_init_nets* **then**
        **foreach** *i in network_batch* **do**
            selected_prompt = rand_prompt;
            generated_network, temperature= `Mutation`;
            prompt_individual = selected_prompt, temperature;
            `add_to_archive`;
        **end**
    **else**
        evolutionary_operator = mutation or crossover;
        **foreach** *i in network_batch* **do**
            **if** *evolutionary_operator == mutation* **then**
                rand_network_from_network_archive;
                from_prompt_archive_get: half_curious_prompt_individuals;
                half_random_prompts_individuals;
                generated_network, temperature=`Mutation`;
                prompt_individual = selected_prompt, temperature;
                all_prompts += prompt_individual;
            **else**
                generated_network = `Crossover`;
                all_networks += generated_network;
            **end**
        **end**
    **end**
    **foreach** *i in all_networks* **do**
        accuracy = `train`;
    **end**
    get_network_with_best_test_accuracy; get_corresponding_prompt_individual;
    `add_to_archive`;
**end**

---

`LLMatic` begins its search with a very basic neural network, inspired by the work of Stanley & Miikkulainen (2002) which suggests that neuroevolution tends to perform better when starting with a small network. In `LLMatic`, we use dual-archive cooperative QD optimization, a method where two separate archives are used to store complementary components that can be combined to solve a given task. The first archive stores neural networks, where the width-to-depth ratio and Floating Point Operations per Second (FLOPS) of a network are the behavioural descriptors. The width-to-depth ratio is a division of the width and the depth of the network. To specify the width, we use the maximum of the output features of all layers, while number of layers is considered to be the depth of the network. FLOPS were chosen over parameter count because FLOPS correlates better with actual time spent training a network Ayala et al. (2017). We call this archive the "network archive". The fitness function for the networks in this archive is defined as the test accuracy of

the network after training. The second archive, called the "prompt archive", contains the prompt and temperature for generating code, which is the behavioural descriptors as well. The selection of prompt and temperature depends on a curiosity score (Cully & Demiris, 2017), which depends on whether the generated network was added to the network archive. The fitness of prompt individuals depends on whether the network was better than the previous generation's best score.

In the first generation , ta simple neural network with one convolutional and one fully connected layer initiates the evolution. A prompt is selected at random to generate an initial batch of networks. These networks are evaluated and an attempt is made to add them to the network archive as a random initialization for MAP-Elites. Concurrently, we mutate the temperature based on the fitness of the network, increasing it if the fitness increases and vice versa. By increasing the temperature, we want the LLM to explore as it is performing better than before. By decreasing the temperature, we want the LLM to exploit and try to achieve better fitness than before. Once we calculate the fitness of the prompt individual, we add the score to a collective prompt fitness score, after which we try to populate the prompt archive. The collective prompt fitness score determines the overall fitness of each individual in the prompt archive as it gives each prompt a fitness score.

Once either of the archives reaches a specified capacity, we introduce training of the neural network and evolutionary operators in the evolution process. With a certain probability at each generation, a decision is made on whether to perform crossover or mutation to produce $N$ new batch of offspring. If a crossover is chosen, we select $N$ random network individuals, locate their closest networks in the archive, and carry out a crossover operation instructed by a prompt. No individual is added to the prompt archive when a crossover is performed. If the mutation operation is selected, we pick the most curious prompt individual and a random network individual. For exploration, we also select random prompts. In both cases, each network is trained for a certain number of epochs and an attempt is made to add the network to the archive. Likewise, a prompt individual is added as previously described. This process continues for a pre-determined number of generations. Algorithm 1 shows the complete search process of `LLMatic` in pseudocode. Refer to supplementary material for pseudocodes on mutation operators, crossover operators, temperature mutation and addition to archives.

## 4    CURATING LLMATIC

To curate `LLMatic`, we use CIFAR-10 (Krizhevsky et al., 2009), a commonly used dataset for NAS (Tan & Le, 2019; Ying et al., 2019). We perform extensive ablation studies to demonstrate that `LLMatic` requires all of the components for the search. Once our algorithm is curated, we extend our experiments to *NAS-bench-201* which is a queryable dataset Mehta et al. (2022). Queryable datasets allows us to search for network architectures without training them.

### 4.1    SETTING UP LLMATIC

**Dataset:** The CIFAR-10 dataset is made up of 60,000 color images, each with a resolution of 32x32 pixels, and divided into 10 categories. The categories are airplane, automobile, bird, cat, deer, dog, frog, horse, ship and truck. The CIFAR-10 dataset is partitioned into five groups for training and one group for testing, each group holding 10,000 images. Each test group consists of an exact count of 1,000 images from each category, selected randomly. The training groups hold the remaining images, which are arranged in random order. As a result, some training groups might contain more images from one category compared to others. Nonetheless, collectively, the training groups have an exact total of 5,000 images from each category.

**Initial Neural Network:** `LLMatic` starts off with a simple neural network with one convolutional layer that takes in 3 input with 3 channels, with $1 \times 1$ kernel size and 1 output channel which connects to a dense layer with 1024 input neurons. Since the size of images in the dataset is $32 \times 32$, with 1 (grayscale) channel, the linear layer will have 1024 hidden neurons. These hidden neurons are connected via another dense layer to 10 output neurons (as we have 10 classes). Rectified Linear Unit (ReLU) (Nair & Hinton, 2010) is the activation function used in all layers. All of our networks are generated in PyTorch (Paszke et al., 2019).

**Generating Neural Networks:** At each generation, we generate a batch of 100 new offspring. Each network generated is trained for 50 epochs. The networks are optimized by stochastic gradient

descent (Bottou, 2010) with the learning rate set to 0.001 and momentum set at 0.9 for all networks. We use Cross Entropy loss as our measure for the fitness of the trained network.

For evolutionary operators, we set a probability of 0.7 for mutation and 0.3 for crossover as after experimentation, we found mutation to create consistently more trainable neural networks. We initialize the temperature parameter (used when sampling the code-generating LLM) to 0.6. For temperature mutation, half of the population is generated by the prompt individual temperature mutated randomly between $-0.1$ to $0.1$. The other half is generated by the temperature obtained from the prompt individual itself. If the fitness of the generated network is better than or equal to the best fitness of the previous generation, we increase the temperature by 0.05 and if it is worse than the best fitness of the previous generation, we decrease it by 0.05. For the crossover operator, we select 10 random networks and find their 2 or 3 nearest neighbours in the network archive to perform crossover. We set the temperature to be 0.7 for network generation.

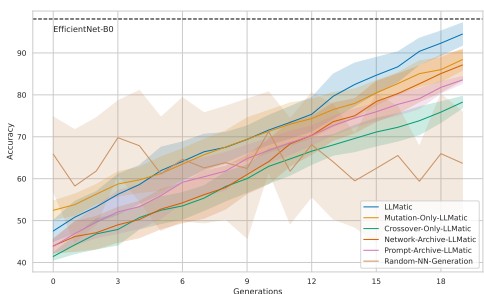

Figure 1: The illustration of the best accuracy per generation for `LLMatic` and all ablation studies. Each experiment is conducted with 10 seeds. The shaded region is the standard deviation while the solid line represents the mean. EfficientNet-B0 is the best-performing EfficientNet on CIFAR-10.

**Quality Diversity Optimization:** For our QD optimization algorithm, we choose a variant of MAP-Elites, called Centroidal Voronoi Tessellation (CVT-MAP-Elites) (Vassiliades et al., 2017), which was created to scale up MAP-Elites and is shown to do so by comparing with other variants of MAP-Elites by Nilsson & Cully (2021). CVT-MAP-Elites automates the creation of the archive by creating $k$ cell centroid locations that spread through the behavioural descriptors space. We use the *pymap_elites*[1] implementation for our experimentation. We use k-d tree (Bentley, 1975) to create and write centroids to the archive and find the nearest neighbors using a Euclidean distance metric (Dokmanic et al., 2015).

For our QD archives, we use 10 niches per dimension, and we have 2 dimensions per archive. We set our number of random initial networks to 10. Random initial networks are needed to be filled in archives before evolutionary operators are introduced. For the network archive, we have the width-to-depth ratio of the network as our first dimension and the FLOPS of the network as the second dimension. The width-to-depth ratio has a lower limit of 0 and an upper limit of 200. The minimum FLOPS is set to 200 MegaFLOPS and the maximum is set to 5 GigaFLOPS. This range is set after experimentation. For the second archive, i.e. the prompt archive, we have the prompt encoded as an integer as the first dimension and temperature as the second dimension. The maximum value of the prompt equals the number of prompts we have, which is 16 in total. The maximum value of temperature is set to 1 as it can never increase beyond that for our LLM. The lower limit for all the dimensions is 0. For the network archive, we simply select a random network while for the prompt archive, we select the most curious prompt individual which depends on the curiosity score. This curiosity score is incremented by 1.0 if the selected prompt adds the generated network to the network archive, decreased by 0.5 if the network is not added, and reduced by 1.0 if the created network is untrainable. If the generated network has better fitness than the previous generation's best network, the collective prompt fitness score for the prompt in the prompt individual is increased by 1, otherwise, it is unchanged. We use prompts that are generalizable to any problem in any domain. Refer to Appendix A for an example of mutation and crossover prompts.

**Code Generating LLM:** We use the pre-trained CodeGen (Nijkamp et al., 2022) LLM to generate neural networks. CodeGen is an autoregressive decoder-only transformer with left-to-right causal masking. CodeGen is first trained on `ThePile` dataset with random initialization and is called CodeGen-NL. CodeGen-Multi is initialized with CodeGen-NL and is trained on `BigQuery` dataset. Lastly, CodeGen-Mono is initialized with CodeGen-Multi and is trained on `BigPython`. CodeGen

---

[1] https://github.com/resibots/pymap_elites

is trained to be in various parameter sizes but we use 6.1 Billion parameter variant of CodeGen-Mono due to computational constraints.

`ThePile` dataset (Gao et al., 2020) is an 825.18 GB English text corpus. Nijkamp et al. (2022) select a subset of the Google `BigQuery` dataset which contains 6 programming languages, namely C, C++, Go, Java, JavaScript, and Python. The authors collected a large amount of permissively licensed Python code from GitHub in October 2021, and named it `BigPython`. The size of `BigPython` is 217.3 GB.

CodeGen-6B has 33 layers and 16 heads with 256 dimensions per head. The context length is 2048 and the batch size is 2 million tokens. Weight decay is set to 0.1. $0.4e^{-4}$ is the learning rate. Warm-up steps are set to $3k$ while total steps for training are 150k.

## 4.2 ABLATION STUDY

As we have many components in `LLMatic`, we choose to do a thorough ablation study to determine the effect of each component on overall performance. The following are the components tested for the ablation study:

- `Network-Archive-LLMatic`: LLMatic with only network archive. To achieve this, we create a prompt individuals population. The population is fixed to 100 individuals starting from creating random individuals. We have only one fitness score for this population, which is calculated as $+1$ if a network is added in the network archive, $-0.5$ if the network is not added and $-1$ if the network is not trainable. After we generate the network, we mutate temperature by adding 0.1 if the network is added in the network archive and $-0.1$ if the network is not added.

- `Prompt-Archive-LLMatic`: LLMatic with only prompt archive. To achieve this, we create a population of networks. The fitness function for the population of networks is accuracy. We keep the population to a 100 individuals. With a similar probability as `LLMatic`, we select mutation or crossover operator. For the crossover operator, we select the individual that is closest to the structure of the selected network. For network similarity, we use cosine similarity and we choose the networks with higher scores. For the mutation operator, similar to `LLMatic` we mutate half of the networks from the most curious prompt individuals and half from random individuals.

- `Mutation-Only-LLMatic`: LLMatic with only mutation operator.

- `Crossover-Only-LLMatic`: LLMatic with only crossover operator.

- `Random-NN-Generation`: Neural network generation without evolution. We generate 100 networks per generation for 20 generations to make it comparable as it is the same number of networks generated per batch in LLMatic. We apply the prompt "Create a neural network that inherits from nn.Module and performs better than the above neural network" and we add the initial network with this prompt.

### 4.2.1 ABLATION RESULTS AND DISCUSSION

In this section, we will discuss the results of the experiments that we set up in the previous section. We first discuss the best loss per generation, illustrated in Figure 1. This will lead our discussion to trainable networks generated by changing the crossover and mutation probabilities (Figure 2). Then we will discuss how archives are illuminated Figure 3. Some of the generated networks are shown in the supplementary material.

Looking at Figure 1, it is clear that each component of `LLMatic` is necessary. `Mutation-Only-LLMatic` and `Network-Archive-LLMatic` are the closest to `LLMatic` which also proves that our choice of giving mutation more probability of being selected is the right one. `Crossover-Only-LLMatic` is understandably the worse as mutation provides more exploration (Ullah et al., 2022). Both operators, mutation and crossover, together give exploration and exploitation abilities to `LLMatic`, which are highly necessary to find high-quality and diverse networks. While `Prompt-Archive-LLMatic` does significantly worse as network archive is an important aspect to find high-performing networks. Both archives together demonstrate competitive results.

We use EfficientNet-B0, which is the state-of-the-art network on CIFAR-10, from Tan & Le (2019) as an indicator of where our algorithm stands. EfficientNet-B0 was searched via methods applied by Tan et al. (2019) and is slightly larger than the original study as they were targeting more FLOPS. The original study required 8000, while `LLMatic` requires 2000 searches to find a competitive network. EfficientNet-B0 was first trained on ImageNet dataset (Deng et al., 2009) and then on CIFAR-10 via transfer learning (Torrey & Shavlik, 2010). This is an advantage for EfficientNet-B0 as ImageNet has many classes and is an order of magnitude larger dataset.

Figure 3 demonstrates how each archive is being filled on average. We can see the prompt archive contains high-performing individuals who have the first few prompts and higher temperatures.

Figure 2: The illustration of how many trainable networks are created in a generation. The total number of networks created is 100 per generation.

Some of the good-performing individuals do have lower temperatures which suggest that sometimes it is good to pick deterministic layers. For network archives, we observe a diversity of high performing networks with respect to both FLOPS and width-to-depth ratio. More than 20 networks are competitive networks in this archive.

To delve into why we choose the probabilities being 0.7 for mutation and 0.3 for crossover, we observe the number of trainable networks generated per generation (see Figure 2). This is to be considered common knowledge that the more working individuals we have, the greater the chance of high-performing individuals. For this purpose, we train `LLMatic` with uniform probabilities, and 0.3 for mutation and 0.7 for crossover. We observe that uniform probabilities are still competitive with the original setting, while increasing the crossover probability makes it worse. The results of these experiments and results of the ablation study for Mutation-Only-LLMatic and Crossover-Only-LLMatic lead us to the conclusion that mutation should be given more probability of being selected.

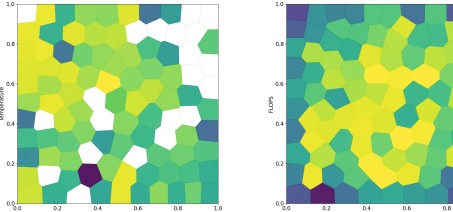

(a) Prompt archive: Prompts encoded as integers on the $x$-axis, normalised to be in range 0-1 for CVT-MAP-Elites as all points are within 0–1. On the $y$-axis, we have the temperature that controls LLMs exploration ability. As 1 is the maximum temperature, there is no need for normalisation.

(b) Network archive: Width-to-depth Ratio on the $x$-axis. The range for the Width-To-Depth ratio is from 0 - 200 normalised to 0-1. On the $y$-axis, we have Floating Point OPerations per Second (FLOPS). We have a range of 200 Mega FLOPS to 5Giga FLOPS. This range is normalised to 0-1 for CVT-MAP-Elites.

Figure 3: An illustration of archives generated by `LLMatic`. We have selected the archive with the median number of cells filled in experiments over 10 seeds. Figure 3a shows prompt archive. Figure 3b shows network archive. The lighter the colour of the filled cell, the better fitness of the individual.

## 5 EXPERIMENTS ON NAS-BENCH-201

### 5.1 DATASET AND BENCHMARK

Next, we extend our experimentations of `LLMatic` on NAS-bench-201 (Dong & Yang, 2020) benchmark, which searches a cell block for a constant neural network structure. The structure is initiated with one 3-by-3 convolution with 16 output channels and a batch normalization layer (Ioffe & Szegedy, 2015). The main body of the skeleton includes three stacks of cells, connected by a residual block. Each cell is stacked 5 times, with the number of output channels as 16, 32 and 64 for the first, second

Table 1: A comparison of test accuracy on NAS-bench-201 `LLMatic` with Λ-DARTS (Movahedi et al., 2022), having the most near-to-optimal result, GENIUS (Zheng et al., 2023), a GPT-4 based neural architecture search algorithm and Random Search, implemented by NAS-bench-201 authors (Dong & Yang, 2020). We provide optimal accuracy for reference, which is the maximum accuracy that can be achieved in NAS-bench-201.

| Method | CIFAR-10 | CIFAR-100 | ImageNet16-120 |
|---|---|---|---|
| Random Search | 93.70±0.36 | 71.04±1.07 | 44.57±1.25 |
| GENIUS | 93.79±0.09 | 70.91±0.72 | 44.96±1.02 |
| Λ-DARTS | 94.36±0.00 | 73.51±0.00 | 46.34±0.00 |
| LLMatic | 94.26±0.13 | 71.62±1.73 | 45.87±0.96 |
| Optimal | 94.47 | 74.17 | 47.33 |

and third stages, respectively. The intermediate residual block is the basic residual block with a stride of 2 (He et al., 2016), which serves to downsample the spatial size and double the channels of an input feature map. The shortcut path in this residual block consists of a 2-by-2 average pooling layer with a stride of 2 and a 1-by-1 convolution. The skeleton ends with a global average pooling layer to flatten the feature map into a feature vector. Classification uses a fully connected layer with a softmax layer to transform the feature vector into the final prediction.

The specified cell within the search domain is depicted as a densely connected directed acyclic graph with four nodes and six edges; here, nodes symbolize feature maps while edges denote operations. There are five possible operations: (1) zeroize, (2) skip connection, (3) 1-by-1 convolution, (4) 3-by-3 convolution, and (5) 3-by-3 average pooling layer. Zeroize drops out the associated edge operation. Given five operations to choose from, the aggregate count of potential search spaces comes to $5^6 = 15625$ cell combinations. Evaluations are carried out on CIFAR10, CIFAR100 (Krizhevsky et al., 2009), and ImageNet16-120 (Chrabaszcz et al., 2017). ImageNet16-120 is a variant of ImageNet dataset (Russakovsky et al., 2015) which is downsampled to 16x16 image sizes and contains the first 120 classes.

## 5.2 RESULTS

To stay consistent with our previous experiments, `LLMatic` searches for 20 generations and 100 cells in a generation. We curate the prompt to cater for a controllable generation by restricting it to the five operations. Refer to Appendix A for an example of how we generate queryable cells. For our network archive, we take minimum and maximum FLOPS as the bounds for the behaviour descriptor.

We compare our results with GPT-4 based NAS algorithm GENIUS (Zheng et al., 2023) as an LLM baseline, Λ-DARTS (Movahedi et al., 2022) as it achieves close to optimal result, where optimal is the maximum test accuracy, and Random Search. As Table 1 indicate, `LLMatic` better results than simple GPT-4 based NAS and close to the state-of-the-art and optimal results.

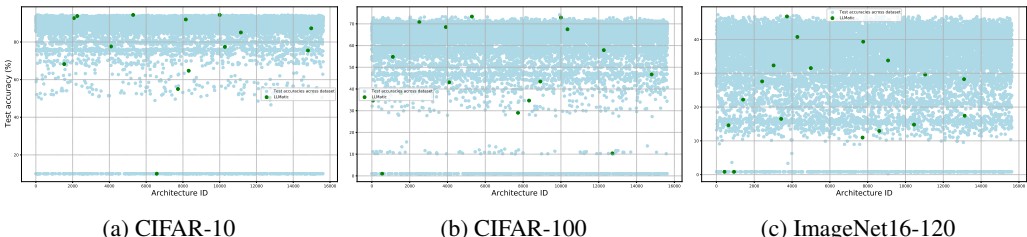

| (a) CIFAR-10 | (b) CIFAR-100 | (c) ImageNet16-120 |

Figure 4: Illustration of test accuracies of all networks across all datasets and best found networks in each generation by `LLMatic`.

Furthermore, in Figure 4 we look into the found networks by `LLMatic` over each generation. We observe not only the near-to-optimal network being found but also the distribution of found networks in the search space. This is due to the procedural nature and exploration capabilities of `LLMatic`

Table 2: Maximum rank achieved by `LLMatic` on each dataset in NAS-bench-201.

| Method | Rank |
|---|---|
| CIFAR-10 | 2 |
| CIFAR-100 | 2 |
| ImageNet16-120 | 11 |

through prompt archive. To demonstrate near-to-optimal networks we look into Table 2 for maximum ranked networks based on test accuracies searched by `LLMatic`.

## 6 CONCLUSION AND FUTURE WORK

To conclude, we present `LLMatic`: a novel Neural Architecture Search (NAS) algorithm that harnesses the power of Large Language Models (LLMs) and Quality-Diversity (QD) optimization algorithms. `LLMatic` successfully finds competitive networks that are diverse in architecture. We show empirically that `LLMatic` can find more than *20* competitive networks in CIFAR-10 and near-to-optimal networks in NAS-bench-201, using only *2000* searches. `LLMatic` decreases the max population size per generation to only *100*. `LLMatic` achieves this while relying on a *6.1B* parameter language model. Furthermore, we show that each component in `LLMatic` is necessary. We do an extensive ablation study and find that `LLMatic` finds the network with the best accuracy among other variants.

`LLMatic` achieves this with many constraints in hand. Firstly, we use CodeGen-6.1B code generation LLM, which is a smaller language model when compared to existing LLMs. This demonstrates how computationally efficient `LLMatic` is, and how much it can unlock the value with a larger language model. Secondly, due to computational resources, we keep our searches to 2000, and still find competitive networks.

In future work, `LLMatic` should be compared to other NAS methods on other computer vision and natural language processing tasks. As neuroevolution is similar to NAS, `LLMatic` needs to be compared to Reinforcement Learning benchmarks as well. With this, `LLMatic` can be used in tasks like Open-ended Learning as well (Nasir et al., 2022).

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

## A  PROMPTS

Prompts used to curate LLMatic:

```
'''"""Add a layer to improve the above network"""'

'''"""Delete a layer to improve the above network"""'

'''"""Improve the above network"""'

'''"""Improve the above network by reducing the size drastically"""'

'''"""Improve the above network by increasing the size drastically"""'

'''"""Add fully connected layer to improve the above network"""'

'''"""Add convolutional layer to improve the above network"""'

'''"""Add pooling layer to improve the above network"""'

'''"""Add residual connection to improve the above network"""'

'''"""Add multiple residual connections to improve the above network"""'

'''"""Add dropout layer to improve the above network"""'

'''"""Add normalization layer to improve the above network"""'

'''"""Add recurrent layer to improve the above network"""'
```

A mutation prompt example:

```
import torch
```

```python
import torch.nn as nn
import torch.nn.functional as F

class Net(nn.Module):
    def __init__(self):
        super().__init__()
        self.conv1 = nn.Conv2d(3, 1, 1)
        self.fc1 = nn.Linear(1024, 10)

    def forward(self, x):
        x = F.relu(self.conv1(x))
        x = torch.flatten(x, 1)
        x = F.relu(self.fc1(x))
        return x

"""Add a layer to improve the above network"""
```

A crossover prompt example:

```python
import torch
import torch.nn as nn
import torch.nn.functional as F

class Net1(nn.Module):
    def __init__(self):
        super().__init__()
        self.conv1 = nn.Conv2d(3, 1, 1)
        self.fc1 = nn.Linear(1024, 10)

    def forward(self, x):
        x = F.relu(self.conv1(x))
        x = torch.flatten(x, 1)
        x = F.relu(self.fc1(x))
        return x

class Net2(nn.Module):
    def __init__(self):
        super().__init__()
        self.conv1 = nn.Conv2d(3, 1, 1)
        self.conv2 = nn.Conv2d(1, 2, 1)
        self.fc1 = nn.Linear(2048, 10)

    def forward(self, x):
        x = F.relu(self.conv1(x))
        x = F.relu(self.conv2(x))
        x = torch.flatten(x, 1)
        x = F.relu(self.fc1(x))
        return x

"""Combine the above two neural networks and create a third neural
    network class that also inherits from nn.Module and performs
    better than the above two neural networks"""
```

Prompts used by LLMatic for NAS-bench-201:

```python
'"""Add ReLU, 3x3 convolutional layer and Batch normalization layer
    to improve the above network"""'

'"""Add ReLU, 1x1 convolutional layer and Batch normalization layer
    to improve the above network"""'

'"""Delete a layer to improve the above network"""'
```

```
’"""Add 3x3 average pooling layer to improve the above network"""’

’"""Add skip connection to improve the above network"""’

’"""Delete a layer to improve the above network and add ReLU, 3x3
    convolutional layer and Batch normalization layer"""’

’"""Delete a layer to improve the above network and add ReLU, 1x1
    convolutional layer and Batch normalization layer"""’

’"""Delete a layer to improve the above network and add 3x3 average
    pooling layer"""’

’"""Delete a layer to improve the above network and add skip
    connection"""’
```

