# OpenReview forum: "LLMatic: Neural Architecture Search via Large Language Models and Quality Diversity Optimization"
_ICLR.cc/2024/Conference — Submitted to ICLR 2024_

### Official Review · Reviewer_za3A · 2023-10-21

**Soundness:** 2 fair
**Presentation:** 1 poor
**Contribution:** 3 good
**Rating:** 3
**Confidence:** 4

**Summary:**

This paper studies an interesting topic regarding how to leverage Large Language Model to generate DNN architecture. The proposed LLMatic method jointly utilizes prompts, QD, two types of evolution operators, i.e., mutation and crossover,  to produce high-performing DNNs. The numerical results on CIFAR10 and NAS-bench-201 show the effectiveness of the proposed approach.

**Strengths:**

After reading the manuscript, I summarize the below strengths.

- The topic of the paper is new and interesting. Leveraging generative models may produce impacts into the NAS field.

- The designed algorithm makes sense that should produce some high-performing DNNs.

**Weaknesses:**

Meanwhile, I have following concerns.

- The presentation of the paper is not satisfactory. Especially the pseudocode, which lacks proper mathematical annotations. Meanwhile, there exist a lot of typos, e.g., 'ta simple' on page 4. The legends in Figures are invisible, etc.

- Again the presentation, the key components in the algorithm such as mutation operators, crossover operators and temperature mutation lacks proper description and explanations. It makes the algorithm unclear.

- This paper looks more like an investigation paper to explore how to use LLM on generating new DNNs, yet lacks sufficiently novel algorithm to guide LLMs generating DNNs of higher fidelity. The rank on ImageNet is significantly lower than CIFARs, which increases my concern regarding the effectiveness of the proposed algorithms on general tasks.

**Questions:**

- What is the search cost of the algorithm?

---

### Official Review · Reviewer_j44t · 2023-10-23

**Soundness:** 3 good
**Presentation:** 1 poor
**Contribution:** 3 good
**Rating:** 5
**Confidence:** 3

**Summary:**

The authors propose an LLM-based NAS algorithm which leverages the coding ability of LLMs in combination with a type of evolutionary search which allows to mutate architectures directly in the code of the architecture rather than on e.g. an abstract graph level. The authors employ QD for their evolutionary search, which results in a wide-range of solutions which can e.g. fulfill different hardware requirements. The authors demonstrate good performance on NAS-Bench-201 using only 2000 queries.

**Strengths:**

- I like the idea of carrying out NAS directly on the code-level instead of an abstract representation such as a DAG, as it makes the search flexible. This also removes the need for a compiler from the abstract model description to the code which can save time and hence lower the barrier to actually use NAS. Joint NAS and HPO may also be expressed in a very natural way in this paradigm.
- Using quality diversity in this paradigm makes sense to me, not just for diverse hardware requirements, but also for good ensembling.
- The ablation experiment nicely illustrates the contribution of each component.

**Weaknesses:**

- Could you run additional experiments on Nas-Bench-101? As the search space is constrained in a different way than 201, I would be interested to see whether your approach can remain within the search space at all times.
- Could you provide optimization trajectories for Nas-Bench-201 along with a baseline such as random search? I am curious how your approach compares with more traditional approaches in terms of any-time performance.
- Overall the visual quality of the paper may be improved. In Figures 2 and 3 the white space around each Figure should reduced. Figure 4 should be changed to a jpg or contain a reduced number of points because it causes rendering issues. Also please increase the fontsize to be at least footnotesize for all figures.
- I find Algorithm 1 to be hard to read. As you don't define each function you use in the pseudo code, you could consider writing short sentences instead.

**Questions:**

- Please check that your references use the published version of a paper (if available) e.g. NAS-Bench-Suite was published at ICLR 2022.

Typos (only minor and non exhaustive, just listing them for completeness):
- 'In the first generation , ta simple neural network'
- '... layer that takes in 3 input(s?) with 3 channels'

---

### Official Review · Reviewer_7vVP · 2023-10-31

**Soundness:** 2 fair
**Presentation:** 2 fair
**Contribution:** 2 fair
**Rating:** 3
**Confidence:** 4

**Summary:**

This paper proposes the use of LLMs' code generation capability to generate network models, along with the use of Quality-Diversity for search. The authors validate the feasibility of the method on multiple datasets and search spaces.

**Strengths:**

- The use of large models' knowledge for network structure generation provides an example of applying large models in practice.

**Weaknesses:**

- While utilizing the code generation capability of large models for network structure generation is a good application, previous works such as GENIUS have already proposed very similar methods and demonstrated prompts. Therefore, the novelty and contribution of this paper appear insufficient.
- The experimental description is not detailed, and the analysis is not sufficiently deep. For example, Table 1 does not report the search cost of this paper compared to related works.
- There are relatively few experimental comparisons, and the comparison with related works is not comprehensive enough.
- After reading this paper, I did not get much insight. Importantly, the assistance of prior knowledge from LLMs in network structure design remains unclear.

**Questions:**

Recommendations:
- Provide a more comprehensive discussion on the novelty and contribution of the proposed method compared to similar existing approaches.
- Elaborate on the experimental setup and provide more detailed analysis of the results, including reporting the search cost and the impact of prior knowledge from large models on network structure design.

---

### Official Review · Reviewer_4jrW · 2023-11-01

**Soundness:** 3 good
**Presentation:** 2 fair
**Contribution:** 3 good
**Rating:** 3
**Confidence:** 4

**Summary:**

This work proposes LLMatic which is a Neural Architecture Search approach using LLMs and quality diversity optimization. This work combines the domain knowledge of code-generating LLMs with a robust quality-diverse search mechanism.The application of LLMs trained for generating architecture code NAS is novel and interesting way to exploit the capabilities of code-gen LLMs. The method is evaluated on CIFAR-10 dataset and the  NAS-bench-201 benchmark.

**Strengths:**

- The application of code-generating LLMs effectively for NAS is very novel. Defining architecture generation as a language/code modelling task is a new way of formulating the neural architecture search problem.
- The presentation of the paper is mostly clear except in some parts (refer to suggestions and questions in the weakness and questions section)
- Evaluation on the NB201 dataset and CIFAR10 dataset fairly exhaustive and well ablated.
- The authors release their code and additional details on the prompts used

**Weaknesses:**

- Evaluation is limited: Currently NAS focuses a lot of transformer-based spaces [1], mobilenet-spaces [2] which are more practical and realistic compared to cell-based spaces. A lot of thee approaches release a surrogate predictor or the supernet itself, to save training times ie the architecture training part in algorithm 1. I recommend the authors evaluate the method on these search spaces too. This is also in my opinion very important to study how llmatic scales across larger architecture definitions (code) and datasets eg: ImageNet.
- Comparison to black box approaches and generative approaches: Since llmatic requires training a lot of architectures (or querying a benchmark multiple times), its search time is more comparable to black-box approaches instead of approaches like lambda-darts. It would be great to add other black-box methods (in addition to random search) to Table 1. Furthermore since the work very much falls in the line of generative NAS a comparison with DiffusionNAG [3] would also be great.
- Clarity: In the current version of the paper in figure 4, I couldn't see the green points referring to llmatic architectures, am I missing something?
- Minor : Page 4 2nd paragraph "In the first generation , ta simple neural network with one convolutional and one fully connected
layer initiates the evolution" -> "In the first generation , take simple neural network with one convolutional and one fully connected
layer initiates the evolution"

[1] Chen, M., Peng, H., Fu, J. and Ling, H., 2021. Autoformer: Searching transformers for visual recognition. In Proceedings of the IEEE/CVF international conference on computer vision (pp. 12270-12280).

[2] Cai, H., Gan, C., Wang, T., Zhang, Z. and Han, S., 2019. Once-for-all: Train one network and specialize it for efficient deployment. arXiv preprint arXiv:1908.09791.

[3] An, S., Lee, H., Jo, J., Lee, S. and Hwang, S.J., 2023. DiffusionNAG: Task-guided Neural Architecture Generation with Diffusion Models. arXiv preprint arXiv:2305.16943.

**Questions:**

- Evaluation: Check points in the weakness (evaluation)
- Clarity : Code generating llms are often imperfect in generating pytorch code. They often introduce bugs and require some human intervention to make a generated code snippet compilable. Did the authors face any such issues? In other works how does a given prompt ensures that an LLM would restrict itself to generate compilable architectures within the NABench 201 search space for eg? The prompt examples provided do not seem to indicate any instructions to the LLM to ensure this
- Could the authors comment on how their work compares to [4] which explores a very similar direction (but with prompt evolution)?
- Reproducibility: Reproducibility of approaches is a major challenge in NAS. Since the work relies on (open-source) LLMs could the authors comment on the reproducibility of their approach?
- Could you report search time comparison across all baselines in table 1?

[4] Chen, A., Dohan, D.M. and So, D.R., 2023. EvoPrompting: Language Models for Code-Level Neural Architecture Search. arXiv preprint arXiv:2302.14838.

---

### Official Review · Reviewer_eaNZ · 2023-11-01

**Soundness:** 2 fair
**Presentation:** 1 poor
**Contribution:** 1 poor
**Rating:** 3
**Confidence:** 5

**Summary:**

The paper introduces LLMatic, a NAS algorithm that combines LLMs with Quality-Diversity (QD) optimization to efficiently generate diverse and high-performing neural network architectures. Utilizing a dual-archive approach, LLMatic reduces the number of candidate evaluations needed, demonstrating competitive performance on benchmark datasets like CIFAR-10 and NAS-bench-201 with just 2,000 candidates. This approach not only leverages the code generation capabilities of LLMs to introduce meaningful variations in network architectures but also ensures robustness and diversity in the solutions through QD algorithms.

**Strengths:**

- the idea of employing LLMs for NAS is novel and aligns with the emerging trend of leveraging these models in diverse fields, considering that LLMs have been applied in domains with structured data formats such as drug design and materials science,

**Weaknesses:**

- LLMatic's performance heavily depends on the LLM's prior exposure to relevant coding patterns and neural network architectures during its training. This dependence could be a potential weakness, as the model might not generalize well to novel or highly specialized architectural search spaces.
- The paper focuses on CIFAR-10 and NAS-bench-201 for performance evaluation. While these are standard benchmarks, the scope of testing could be broadened to include more diverse and challenging datasets. This would provide a more comprehensive understanding of LLMatic's capabilities and potential limitations, especially in real-world scenarios or more complex tasks.
- The paper could benefit from ablation studies to understand the contribution of each component of LLMatic to its overall performance.
- The writing should improve.

**Questions:**

- What's the intuition behind the proposed method? How can we prove that LLMs have the prior knowledge about NAS tasks as described in the paper?
- The input sequence length of current LLMs is limited, but the number of networks evaluated in one experiment exceeds 2000. Is this a major problem, and how do you address it? Or would you consider incorporating the search history into the prompt to assist NAS?
- Why randomly pick a prompt as an action at each step? Why not design prompts that allow LLMs to choose the action by themselves?

---

### Meta-Review · Area_Chair_oQXt · 2023-11-24

**Metareview:**

This paper received 5 reviews, all giving rejection scores. The authors did not rebut.

**Justification For Why Not Higher Score:**

Nobody gave an acceptance score; the authors did not rebut.

**Justification For Why Not Lower Score:**

N/A

---

### Decision · Program_Chairs · 2024-01-16

Reject